# Reading the family: A constructivist grounded theory on organ donation conversations

**Lissette Avilés**📧*

Nursing Studies Department, University of Edinburgh, Edinburgh, Scotland, United Kingdom

* lissette.aviles@ed.ac.uk

**Data Availability Statement:** Thank you for the opportunity to clarify this point. Indeed, the dataset will be available upon request because it contains potentially identifying or sensitive patient/institutional information. This implied that due to the characteristic of the sample, participant could

## Abstract

Approaching families to discuss deceased organ donation authorisation is considered one of the central stages of the organ donation process. In many countries, specialist nurses lead the organ donation process, approach and support families. However, how these encounters occur is not yet fully understood. A constructivist grounded theory methodology was used to conceptualise the process of approaching families from the perspectives of healthcare professionals and families. Data collected included six months of observations across two large hospitals in Chile, documents, interviews and focus groups with 71 participants including healthcare professionals involved in the organ donation process, and bereaved families who were approached for organ donation conversations. The theory Reading the family was developed to explain the relational process of how nurse organ donor coordinators approach families to negotiate organ donation authorization. It explains the sophisticated and skilled process of accessing, assessing and managing family's emotions to negotiate organ donation authorization as a family unit. The theory has two dimensions, indirect and direct, which refers to when and how nurses read families emotions to lead organ donation conversations and support bereaved families' decision-making process. These dimensions critically depend on the clinicians set of beliefs and communication processes. Understanding the complexities of approaching families is essential for practice and policymaking, particularly when there is a trend towards individual decision-making instead of understanding organ donation as a family affair. Reading the family could be eventually applied in other contexts and situations that involve navigating difficult conversations, and therefore, further research is needed and suggested to assess the feasibility of its application.

## Introduction

The organ donation process entails a series of complex interventions to help people through transplantation, and approaching families is considered one of the most relevant stages to obtain organ donation authorization [1–3]. Approaching families consists of the process in which trained healthcare professionals speak to the family of a potential organ donor about organ donation authorisation.

be easily identifiable. As a result, data were managed as strictly confidential but stored in the platform DataVault of my institution, the University of Edinburgh, which contains a DOI and now provided. Apologies for the confusion before. DOI: https://doi.org/10.7488/8f7652c6-5699-423d-894f-3241fb863029 The dataset is securely kept in Data Vault and ca be requested following University of Edinburgh Data Sharing Agreement. The Research Ethics Committee at the University of Edinburgh reviewed and approved the study, which can be also contacted for further information at hiss.ethics@ed.ac.uk.

**Funding:** The research was funded by the Agencia Nacional de Investigacion y Desarrollo, Becas Chile - PhD scholarship, Chile (Folio N. 72160141). The funder had no role in study design, data collection and analysis, decision to publish, or preparation of the manuscript.

**Competing interests:** The author has declared that no competing interests exist.

Organ donation conversations after brainstem death occur when families are dealing with sudden and unexpected loss [4, 5]. Strategies to help professionals to conduct the approach have been described in guidelines [6–8], and simulated courses. Professionals who approach families include physicians, social workers, psychologists and nurses. In countries such as the United Kingdom and Chile, the role of approaching families is carried out only by specialist organ donation nurses, leading this specialised area of nursing through training, communication strategies and skills [9, 10].

Understanding the process of approaching families is paramount when social narratives indicate that family refusal constitutes one of the main drawbacks of the organ donation process [11, 12]. As a result, extensive literature has focused on families' consent and healthcare professionals' skills to support families' needs and the process effectively [13–15]. The optimal moment to approach and discuss organ donation authorization is also discussed in the literature. Separating the communication of the death from the organ donation conversations, also called decoupling, could yield more consents [16], yet other studies suggest otherwise. López et al. [17] analysed the family organ donation decision-making process in sixteen Spanish hospitals, including 421 cases of which 80% resulted in organ donation. Amongst the findings, the authors found no difference in the family's consent of organ donation regardless of whether the conversation occurred separate or in the same conversation of death notification. Also, the highest predictor to consent donation was knowing the deceased's wishes [18].

Cultural differences could play a role when different professions are perceived as more knowledgeable or trusted, which might also depend on organ donation practices and systems amongst countries. For instance, while Poland and Spain reported higher rates of consent by physicians [19, 20], in the UK, more positive outcomes were obtained by nurse organ donor teams [21, 22]. However, it has been described higher consent solely when professionals introduced themselves as the transplant coordinators [17]. The role of these specialist professionals and particularly nurses has drawn further attention in the literature as key agents in the process [23].

The analysis of nurse-family interactions in the context of organ donation is essential when the discussion of organ donation authorisation after the brainstem death of an individual involves family care. How healthcare professionals and families interact and navigate the organ donation process is lacking as well as how families deal with these encounters in the context of their grief. Few studies have examined how these encounters occur internationally [24, 25] and even limited evidence was found in the Latin American context that can inform clinical practice. In Chile, cadaveric donation is possible only after brainstem death and approaching families is carried out by specialist organ donation nurses, drawing recent research attention [26, 27]. Chilean nurses are trained as generalist nurses, undertaking a 5-year bachelor university degree, which also provides professional registration and hence the minimal requirement to work in these roles. In terms of specialist organ donation training, most organ donation nurses receive a 40-hour course delivered by the Chilean National Organ Donation Coordination before commencing their role [27]. In terms of recent statistics, Chile reported 9.6 p.m.p deceased donation rate in 2022, while no recent statistics related to family refusal are published since 2018 around 50% [28]. As a result, the development of theoretical understanding of approaching families is required to better understand the process and support both groups, nurses and families, better.

This study theorises the process of approaching families for organ donation, exploring the care practices, guidelines and healthcare professionals' experiences when involved in the process and bereaved families who were approach for organ donation conversations in Chile.

## Methods

### Study design

A constructivist grounded theory was employed to conceptualise approaching families through an on-going and iterative data collection and analysis, using induction, comparative methods, theoretical sampling, memo writing, and abduction [29]. The constructivist stance acknowledges researcher's role (LA) and positionality in the co-construction of data and theory. The study was reported using the Consolidated Criteria for Reporting Qualitative Research (COREQ) checklist (S1 Checklist).

### Participants recruitment and eligibility

Participants recruitment and data collection were carried out face-to-face between 2017 and 2019 at two large public hospitals in Chile (anonymised as C1 and C2) from 8th November 2017 to 20th June 2019. Data collection began at C1 using purposive sampling strategies and then moved to C2 iteratively and concurrently to compare experiences and practices using constant comparative methods and theoretical sampling. This also ensured transferability beyond one setting. Inclusion criteria included healthcare professionals with experiences supporting families throughout the organ donation process, and families who had been approached for organ donation authorisation, between six months and two years after the experience. As a result, adult donors and non-donors' families participated in the study, who were recruited employing a family subsystems approach [30]. More details of the recruitment process of families have been already published [31]. A thorough informed consent process was conducted, both verbal and written, before data collection started.

### Data collection

Data collection and analysis were carried out by (LA), while conducting her PhD study. A Chilean nurse scholar and qualitative researcher who had studied the organ donation process for more than a decade. Her background as a critical care and dialysis nurse was known in both settings, facilitating access and trust. Four data collection methods were employed, a) participant observation, b) documents, c) interviews and d) focus groups. They allowed comparison of actions, experiences and processes, helping the theorising process [32].

Using purposive sampling, the data collection started with participant observation which began after the organ donation team in C1 agreed to participate and be observed in their daily work. Observations were guided by the question, how is approaching the families for organ donation experienced? Considerations when observing included additional verbal consent before every observation making sure my presence was not disturbing or might affect families' interactions. When observing approaching families, the focus was on organ donation teams instead of families, and hence LA was introduced as shadowing the team. LA did not interact directly with families or participated in these conversations beyond her presence, which was considered appropriate and culturally sounding to the context. Having other staff accompany the teams during this conversation for training was a common practice in C1 and hence my presence was not perceived as an issue.

Handwritten field notes were later transcribed to Word files, and documents such as guidelines, statistics, and forms used by the teams were collated and transformed into PDF files for analysis. Observations and documents informed interviews with healthcare professionals, exploring their experiences while encountering families, which were also explored collectively using focus groups (S1 Appendix). Each family subsystem was interviewed as a unit, separately from healthcare professionals, exploring their family experience when being approached to

discuss organ donation authorisation. Face-to-face interviews and focus groups were audio-recorded with an encrypted device and transcribed verbatim, with an average duration of 71 minutes each. Repeated interviews were not necessary on any occasion, transcripts were not returned to participants as experiences were analysed iteratively using grounded theory strategies. All data were securely stored in the DataStore Service at the University of Edinburgh for data protection.

## Data analysis

Data were systematically collected and analysed using grounded theory practices and strategies [29]. Constant comparative analysis of data uncovered how approaching families for organ donation was a familial and social process, where interactions between organ donation nurses and families were an essential part of meaning and interpretation.

Concurrent analysis refined data collection, interviews and focus groups, moving to C2 site to compare experiences and practices. Line by line coding and gerunds were used to focus on processes and initial coding [29, 32]. Two levels of analysis, comparing what individuals and groups helped to develop codes, focusing on actions, processes and meanings, which progressed in the development of focused codes and then clustered into categories. The iterative analytical process aided by memos informed new observations and theoretical sampling. A detailed analysis process was developed, tracking the theorising process, which was managed, stored and analysed using QSR-NVivo 12.

## Theorising process

Theorising involved constantly questioning the data against theories or concepts, and concepts to data [33]. In this study, from the inductive analytical step of line-by-line coding, towards the end of focused coding and development of categories, data and participants' experiences were theorised employing abduction to theoretically explain and make sense of the data [29, 32].

During the analysis process, it became evident that emotions were essential during the approach and for each participants' experiences. Emotions as dialogical and social entity were thoroughly explored from the perspective of sociology of emotions, psychology and nursing [34, 35]. Solomon [36] refers to emotions as the 'wisdom of the heart' (p.2), or a set of skills to give our lives meaning, and a result, emotions can be purposive strategies to navigate in the world. How organ donation nurses develop these skills and strategies based on their beliefs system was explored [37]. Although the core category, *Reading the family*, was identified early in the study explaining how emotions mediate encounters, dialogue, support and needs for both groups. It was only when the experiences of organ donation nurses and families were sufficiently explained by data, the core category was fully conceptualised and data collection stopped [38]. See Tables 1 and 2 for data set and participants respectively; tables already published [26, 31]. Also, Table 3 succinctly demonstrates the analysis process.

## Rigour and trustworthiness

Grounded theory methodology's rigour criteria were employed in this study [29, 39]. Credibility and resonance were ensured by participants' experiences leading the enquiry process. Gerunds and in-vivo codes were part of the analysis to focus on processes and express explicit and implicit meanings. Workability and usefulness for nursing practice were addressed through the inclusion of two research sites. A systematic reflexivity process was carried out by (LA), recorded in the research journal and regularly discussed in supervisory meetings. Interactions, relationships and processes were theoretically examined and links within data were logically explained. Once the core category was sufficiently explained, data collection stopped using the

**Table 1. Data set.**

| *Data* | |
|---|---|
| Observations | 297 hours |
| Documents | 80 |
| Interviews | 27 |
| | (18) Health professionals |
| | (9) Family members |
| Focus Groups | 14 |
| | (11) Health professionals |
| | (3) Family members |
| *Total Participants* | 71 |
| | (51) Health professionals |
| | (20) Family members |
| *Family approaches* | 6 |
| Family encounters | 21 |
| Donation authorisation | |
| Yes | 3 |
| No | 3 |

**Table 2. Participants' demographics (n = 71).**

| Healthcare professionals n = 51 | | Family members n = 20 | |
|---|---|---|---|
| | | Family units = 14 | |
| Female | 30 | Female | 13 |
| Male | 21 | Male | 7 |
| *Age* | | *Age* | |
| 18–30 years | 10 | 18–30 years | 3 |
| 31–40 years | 24 | 31–45 years | 4 |
| 41–50 years | 7 | 46–60 years | 8 |
| 51–60 years | 9 | 61–70 years | 1 |
| No answer | 1 | >70 years | 4 |
| *Profession* | | *Civil Status* | |
| Nurse | 32 | Married | 10 |
| Physician | 7 | Widow/widower | 5 |
| Psychologist | 7 | Single | 4 |
| Kinesiologist | 2 | Divorced | 1 |
| Nursing Assistant | 3 | | |
| *Professional Experience* | | *Educational Background* | |
| 0–2 years | 5 | Primary School | 6 |
| 2–8 years | 14 | High School | 8 |
| 8–15 years | 16 | Higher Education | 6 |
| >15 years | 16 | | |
| *Organ Donation Experience* | | *Kindship of the Potential Organ Donor* | |
| 0–2 years | 19 | Husband | 2 |
| 2–8 years | 18 | Mother | 6 |
| 8–15 years | 7 | Father | 2 |
| >15 years | 7 | Daughter | 3 |
| | | Brother | 3 |
| | | Sister | 2 |
| | | Son | 2 |

**Table 3. Coding process to develop reading the families.**

| Data | Initial Codes (n = 25) | My Theoretical links | Focused code (n = 4) | Category |
|---|---|---|---|---|
| Valeria: I think it one of most difficult part [..] It is like to see the person in front of you to make them understand the situation (Interview Family member) | Approaching families | Compressed intimacy by Lois? Can they read the family? How? Emotional clues? Belief system? | Reading emotions Dimensions Negotiating consent Approaching families | Reading the family* Dimensions: Direct Indirect (*)To fully understand the core, nurse coordinators and families' experiences conceptualisations were needed |
| Because it is te capacity to see, 'ok, now I'll ask about donation', or 'not now because they are in denial', 'I think.. and so on'. . . This is the most difficult aspect of it (Coordinator 3, Interview) | | | | |
| "The importance of a bad news delivery technique is the communication process, breaking the unilateral communication flow, which makes the patients and family' communication invisible" (Document, MINSAL2017, p. 28) | | | | |
| Coordinator 5:. . .Well, it is like the creation of bond.., LA: Do you think that you manged to create a bond with the families? Coordinator 4: Yeah, yeah . . .a relationship which is very moving because they trust you.. Coordinator 5: It is like a caring relationship, is it? (FG HCP 8) (FGHCP = Focus Groups Healthcare professionals) | | | | |

principle of theoretical sufficiency; data provided enough adequacy and in-depth theoretical understanding of the core process [38].

To explicitly demonstrate the process of abduction while theorising and developing the grounded the findings and discussion sections are presented together. By integrating them, clear and logic connections between data and theoretical literature are directly illustrated [32, 40], and issues related to rigour are addressed [29, 41].

## Ethical considerations

The study received ethical approval from four ethics committees, three in Chile from the relevant National Health Services (AE N043/2017—N994/2017—N5459), and one in the United Kingdom from the School of Health and Social Sciences Research Ethics Committee (NURS028) at University of Edinburgh. All participants provided verbal and written informed consents before data collection began, including regular verbal consent before observations.

Emotional distress was expected due to investigating end of life experiences, yet bereaved families reported feeling empowerment and saw the interview as an opportunity to ask questions, talk, be heard, discuss and contribute to society. As an experienced researcher interviewing bereaved families, rapport developed with families also contributed to minimising the risks of emotional distress, which was acknowledged by participants [42]. Follow-up was employed, including post-communication to check on any issues, questions and provide information about local support. Despite the study offering psychological referral, it was not used nor solicited. Emotional distress was also considered for the leading investigator, who engaged in clinical supervision throughout the research process.

## Results and discussion

### Reading the family

*Reading the family* was developed as the core category to explain the relational process of approaching families of potential organ donors. To explicitly demonstrate the theorising

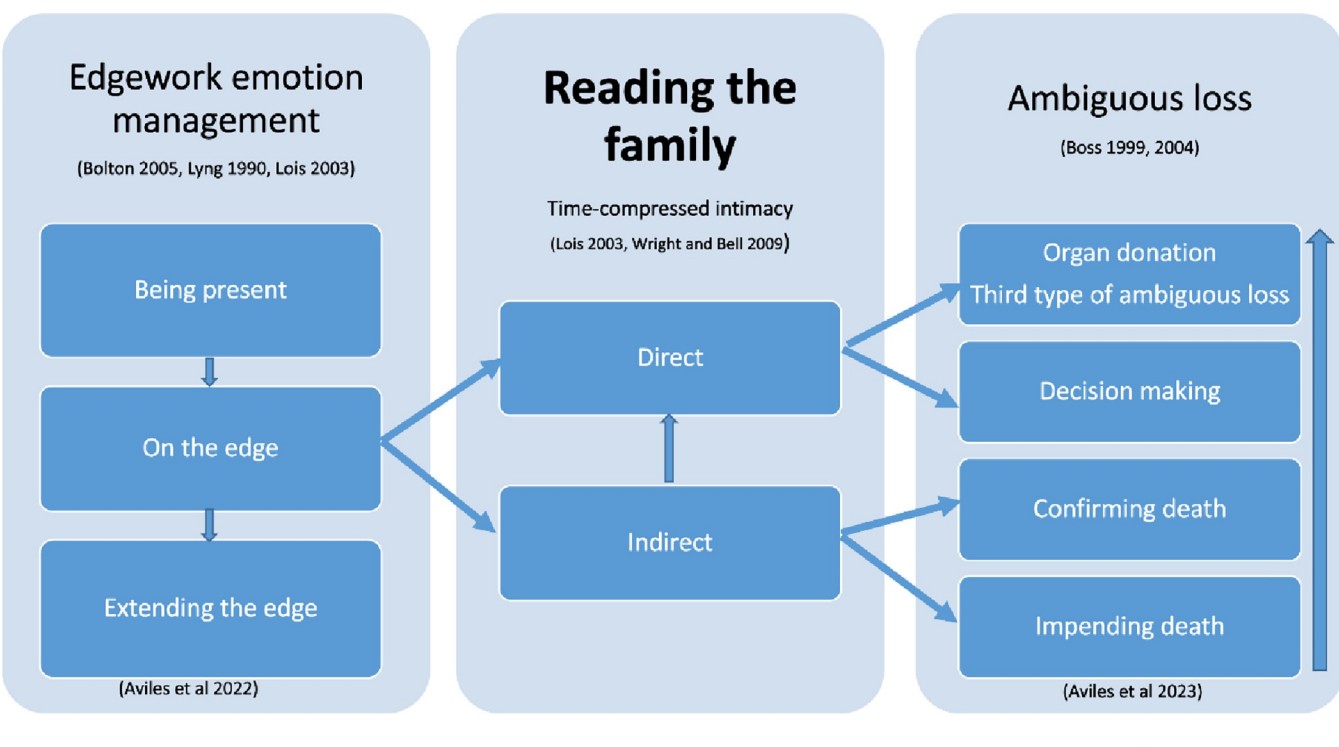

**Fig 1. Reading the family.**

process, findings and discussion sections are presented together and clear links between data and theoretical literature are illustrated.

*Reading the family* explains the sophisticated and skilled process of accessing, assessing and managing family's emotions to negotiate organ donation authorization [43]. This core process has two dimensions that explain how interactions between coordinators and families occur, which are temporal and defined by coordinators beliefs and social norms [37, 44] (See Fig 1).

In Chile, specialist organ donation nurses, also called nurse coordinators, approach families to negotiate organ donation authorisation as a family unit, and as a result, reading families' emotions constitute at its core a nurse-family interaction. Data from observations, documents, interviews and focus groups are employed and presented indented, using anonymised and fictional names. Nurse coordinators in this study were a heterogenous groups in terms of gender and years of experience. For example, two out of the six coordinators were male, yet to ensure their anonymity, quotes and identification are using 'they' and 'the coordinator'.

Discussing organ donation was always perceived conflictive by all participants, healthcare professionals and families alike. The potential disruption of families' experience of loss and the emotional reactions associated with it were acknowledged as the most challenging aspect of the process.

ICU Nurse 12: For me. . .I wonder how colleagues prepare the interview with families.

ICU Nurse 14: It is not easy at all!

ICU Nurse 13: It is not easy at all to speak with a family after the doctor has said 'your family member. . . has passed'; there is no, their brain must stop working. . .

(Focus Group 11, C2)

Another group of participants reflected.

NC staff 1: The training [of organ donor coordinators] consists of enabling them to talk to the family under certain limits. In other words, they need to feel. They won't deliver therapy or alleviate the suffering. Is about humanised communication.

NC staff 2: I mean, the team feels! It is to make it explicit: 'I feel, I'm feeling anxious'. So, how they can facilitate that process, using the family interaction as a resource. It's my process but also the family's.

(Focus Group 2)

Participants consistently described how emotions would run high during family conversations, and the challenges of managing these difficult and sensible conversations in a way that families and nurse coordinators would cope. Articulating the role of emotions to engage and support families in a meaningful manner but also as a resource and skill require to share a scenario, a moment, an interactional space and situation. One nurse coordinator recalls a couple of approaches.

They [the family] had talked about it [organ donation]. It was the father and the husband, and they say 'yes!' straight away. We didn't make any click because they had their decision, and it was an excellent facilitator. In contrast, with another family, yeah, we got the click emotionally. I mean, well, this is a personal matter. She [the mother of the donor] realised that I'm a human being as well. It was very healing for both to weep, for her and for me. We felt very good, though. After that, I met her. . . . I met the lady again (whispering and breathing deeply), and we recognised each other; we hug each other. We didn't say anything, but. . .we both knew that we got our click together at that moment (Emphasis added) (Interview, Nurse coordinator 4, C1)

The nurse coordinator recalls the significance of having this 'click' with the donor's mother. The 'click' constitutes a bond that can connect nurse coordinators and families in an emotional level. Lois [44] described compressed intimacy as the unusual dynamics developed between strangers in crisis that allow working together and managing emotions. In this study, the time for approaching families is limited, lasting from an hour up to a couple of days or a week at maximum. As a result, in this study, the concept used is time-compressed intimacy to describe the meaningful emotional connection between nurse coordinators and families.

Although the time-compressed intimacy could be associated with consenting families, also non-donor families referred to have experienced it. Nora lost her daughter due to a haemorrhagic stroke and refused donation, remembering her experiences, she says.

They let me stay with my daughter. Although it wasn't allowed, I was with her and my two sons all the time. They let everybody in, by two, three of them, everybody. I never moved away from my daughter. They were so respectful to us. . .and the nurse who wept with me. . .It was priceless (trembling voice). She hugged me and wept with me (Interview, Family 10, C2)

Developing time-compressed intimacy involves sharing feelings, dynamics, a dialogue and connection that can remain in some families as precious memories of caring and compassion.

Using the click analogy, Nora got her click with the nurse who wept with her. The need to be supported and understood beyond the clinical role and allowing a person-to-person moment was highly appreciated by families, who felt looked after and cared for. The time-compressed intimacy constitutes a condition of *Reading the family*.

How and when nurse coordinators develop this time-compressed intimacy to read families' emotions depend on teams' beliefs [37] and social norms, leading to *Reading the family directly or indirectly*. If nurse coordinators and hospital staff believe that families can only be approached after the confirmation of brain-stem death, reading the family begins indirectly.

**Reading the family indirectly.**    *Reading the family* indirectly involves assessing family emotions through others. Information of how the family is feeling is passed on to nurse coordinators, based on staff perceptions and staff-family interactions. Liaison with other professionals becomes essential to gather information related to family members' understanding of the situation, emotional state, and needs.

Nurse coordinators' motivations to approach the families after death confirmation obeys to two main aspects, coordinators' beliefs regarding what is appropriate [37] and international recommendations [6, 7]. One of the field notes captured.

> Talking about how they do approach families, one of them says, 'never should be done before death confirmation, and staff might call us to do so'. They emphasise that the approach must be done when the family feels ready (Emphasis added). (Field note, March 2018, C2)

Approaching families of potential organ donors after brainstem death confirmation constituted the gold standard and suggested approach [2, 14, 45]. In other words, nurse coordinators, who follow the standard, begin *Reading the family* indirectly. Through others, nurse coordinator would define the best moment to approach the family based on staff's perceptions of 'when the family feels ready'. A Norwegian study reported the same expression after observing and interviewing 32 ICU professionals involved in approaching families for organ donation. The ideal timing to approach families for organ donation authorisation is "when the family was ready"; a timing that aims to avoid further family distress [46: p.206].

However, this information relies on others' perceptions of families' emotions and understanding of the situation. One of the organ donation teams points out.

> Nurse coordinator 5: I think that. . .the most important part before the approach is to know family dynamics, how they were told about the bad news and the impending death. The communication with the bedside team is essential. So, we can decide to whom to talk to, who of us is going, who are the more resistant to the situation and how to deal with them. It's a preparation, to see from outside the family, how they are dealing with the situation. I make an idea of what is going on.
>
> (silence 5")
>
> Nurse coordinator 6: I completely agree with my colleague. It's seeking, getting bits. Now it's difficult because of the space, we can't see the family easily.
>
> Nurse coordinator 7: As my two colleagues say, we visualise some stuff but from outside still. Therefore, there are so much data that are observed by a third party. Nobody could tell, 'the family is now feeling pain, sorrow'. On the other hand, I think, how are my emotions now to tune in and empathise with them? I feel that we are blind at that moment. [..] I am very concerned about the lack of information. People get limited info regarding the

hospitalisation process, and I got a lot of information. So, I check this first to feel calm and do what I suppose to do.

(Focus Group 8, C2)

Accurate assessment can lead to successful interactions, encounters, communication, support and therefore actions. Nurse coordinator 7 clearly also expresses the link between edge-work emotion management [26] and *Reading the family* when they say, 'how are my emotions now to tune in and empathise with them? I feel that we are blind at that moment', highlighting the challenge of *Reading the family* indirectly.

Another critical point is the family's process of understanding the impending death, particularly the understanding of brainstem death. Informing the imminent death of an individual is recognised as the most difficult aspect of communication [47, 48]. Exploring the experiences of ICU staff, two ICU residents explain.

Physician 5: I mean, they get information, but it isn't clear because they [the family] expect to come to the ICU to get their family member better and it isn't the case.

Physician 4: Certainly, the family isn't told, 'your family member is going to ICU to certify brainstem death' to see if he can help someone, or whatever you wanna say. They know that the patient is pretty poorly, so you go and tell them, 'we're going to ICU'. . .I mean, we, who are more straightforward, say, 'to certify their death'.

Physician 5: Unfortunately, I feel that to some extent we foster this ignorance in the organ donation process.

Physician 4: Indeed. Plus, not all the medics have the technical capabilities or character. Many say, 'it's getting better'. Nothing! 'The sodium improved'. They can't say, 'sorry, there isn't anything else that we can do'. I mean, explaining what is going on. 'Sorry. We can't do more'.

(Focus Group 7, ICU physicians, C2)

Often clinicians wrestle to inform the impending death to families in shock. Davis [49] points out that staff can struggle to manage difficult conversations, which can lead to either evasion or dissimulation. In other words, although clinicians know the prognosis of potential organ donors, it is challenging to talk about impending death to the family openly. Regarding this aspect, an experienced ICU resident said.

Physician 7: It's a physician's responsibility to give the info, nobody else's and it needs to be like this. I mean, there is training in how to deliver bad news, I've done some short courses of it. However! Reading about it is a very different story to talk to the family. When going to the host room to give the info and you see families' faces, obviously. . .Sometimes you forget everything you've learnt.

Feelings permeate interactions, meaning and behaviour, especially when disclosing the impending death of a person. Pattison et al. [50] describe the emotional costs for critical care staff when communicating end-of-life transitions, affecting confidence and self-efficiency. Not surprisingly, one of the strongest recommendations for clinicians is to improve communication skills [51]. Rigid social norms about delivering information to families and managing emotions among staff also influence nurse coordinators' beliefs and practice. As a result, *Reading the family* tends to be in the initial stage indirectly, following the guidelines and gathering

information from critical care staff. One of the centres had a psychological team that provided emotional and psychological support for families in crisis. Exploring psychologists' experiences, the group mentioned.

> Psychologist 3: First, the coordinator asks, who the family is, how many they are, how they are feeling, how they are dealing with everything and collect the info. They gather the info based on the relationship that we had established with the family, indeed.

> Psychologist 4: We participate in the previous stage, working with the traumatic grief because the circumstances are sudden, unexpected [..]

> Psychologist 5: [..] From my experience, I don't see a formal link, a protocol, something between us and the procurement unit. [..] Therefore, I think that could be. . .an opportunity, I don't know, perhaps we haven't talked about it. . .maybe the role that we could play on this.

> (Focus Group 10, C2)

The challenges are numerous, and cooperation is essential. Encountering families in shock and emotionally overwhelmed is often experienced by psychologists at the hospitals who support families going through illness and trauma. As the group mentioned, the rapport developed between psychologists and families provides valuable information that helps nurse coordinators to prepare the approach by *Reading the family* indirectly. Although cooperation channels are not formally established between teams, participants acknowledged collaborative work opportunities.

> Nurse coordinator 7: I think that we need to work more on the process. I think that is too sudden! Because there is <u>no</u> information, a gap between they got the news [the family] and talking to us (Emphasis added by the participant). In my opinion, we need to be clearer. Why do I think that prep[aration] is important? Because I could prepare some aspects of how the family assimilates the news. We got the psychologists; they help us in that phase.

When the process begins through *Reading the family* indirectly, the essential space of emotional dialogue is missed. Nurse coordinators cannot sense the family and how they feel in the experience of death and loss. The ability to assess families' emotions is vital to understand how families experience the dying transition [31] and introduce the discussion about organ donation. Candiotto [52] stresses the essential role of emotions in knowledge building. She proposes that emotions are experienced in the intersubjective relationships that are the result of the affective relationship with the world. Valeria, who lost her daughter due to a brain-injury, reflects.

> Valeria: This interview is without doubts the most complicated aspect of it. Because it's someone like you [referring to interviewer] doing an approach. They're prepared, they must have a technique to do it. Accomplish some, some sort of qualification of . . .empathy, because if the person is doing a good job strictly speaking, right, information. But if they lack closeness but don't have. . .

> I think that these people, the nurse should be special! It can't be anybody! No, because it fills the qualification of. . .of being a good professional, of being excellent. . .They have to achieve some requirements that go a bit further than that.

> LA: Did you feel it from that person?

Valeria: Eh. . . no. I. . .I said to you before, I can't say anything pro or against, but I felt that was just an interview (Emphasis made by the participant).

Families recognise the complexity of these encounters and the discussion of organ donation authorisation. How the approach is performed also involves understanding families' expectations. Families do not always feel cared or understood by professionals, and the notion of closeness and statements such as they should 'truly care' were repetitive during the interviews. The families highlight the need for special qualities related, which could be the development of a compressed intimacy [44]. The social bond here is critical to make the process meaningful, the time-compressed intimacy. *Reading the family indirectly* can miss precious time to develop an emotional dialogue; the 'click', yet it can be the first step for many organ donor teams before encountering families and beginning the conversations about organ donation authorisation.

**Reading the family directly.**    *Reading the family* directly involves the face-to-face encounter between nurse coordinators and families, allowing the development of time-compressed intimacy. These direct encounters can start from the beginning of the hospitalisation process or at later time, when brainstem death is certified. Challenging the guidelines, some organ donation teams develop an early time-compressed intimacy throughout the hospitalisation process. The decision to do so, it depends on nurse coordinators' beliefs [37] and flexible social norms regarding communication processes. Exploring nurse coordinators' motivation to an early time-compressed intimacy, they say.

LA: You approached the family from the beginning of hospitalisation. Why do you do it in this way?

Nurse coordinator 3: Trust. We build trust with the family because we can see them over and over again, and they can see that we do everything for the patient. It is what family is looking for, to know that 'we are doing everything'. We give information, explain procedures. We became exclusive to them. They are grateful.

Nurse coordinator 2: They appreciate it. The family is very grateful because we were there for them. In the A&E they can go in briefly to get the info, 'your family member is poorly'. Instead, we keep them informed all the time!

Nurse coordinator 1: Plus, it helps us as well, I think. We are confident because we know the family in advance. You know who they are; we speak with them. Instead, if someone would come here and say, 'I need to talk to you about the organs!' For me, it would be very uncomfortable.

Coordinator 2: You wouldn't know who you are talking to.

(Focus Group 6, C1)

Acknowledging the sensitive topic to discuss, organ donation authorisation, nurse coordinators believe that trust can help conversations. Early encounters provide knowledge regarding the family and how they are coping with the unexpected situation. To create a time-compressed intimacy, it requires a specialised skill, "letting families set the tone and define the situation" [43: p.133]. By developing a time-compressed intimacy, nurse coordinators let the family to set the tone and nurse coordinators can sense by themselves family's understanding and experience of loss without intermediaries. However, it brings the challenges of observing families suffering side-by-side. One observation describes the impact of observing and feeling families' suffering.

We saw the man kissing her wife, crying silently, hugging her in agony, with a sense of disbelief of what was happening. The sight lost. . . She was a healthy woman two days ago. He finished his farewell and hesitantly head to the entrance to leave the unit. He looked at us touching his face with his hand, hiding, and drying his tears, saying without words 'I cannot speak, goodbye'. I realised everybody was watching; it was heart-breaking. It felt we were trapped in a bubble of suffering; staff were silent, some lowering their sad faces, the coordinator looked at me affected (Field note, 17th November 2017, C1)

In an early time-compressed intimacy, nurse coordinators are exposed to heart-breaking family moments due to a close involvement with the patient and family story. This closeness permits nurse coordinators to provide the opportunity of assisting the family, sometimes with physical demonstrations of affection such as holding hands, hugging and mediating visiting concessions. Early time-compressed intimacy also permits to identify family members, family dynamics, and information gaps accurately.

Coordinators introduce themselves initially as 'nurses' and not directly as organ donation teams. Although this practice was suggested by international guidelines [6, 7], recent studies have contested this recommendation [17]. One of the nurse coordinators reflected.

Nurse coordinator 2: I've commented with my colleague about how the family isn't aware of our job, or they've never said anything. Because they see us at the A&E, in the ICU, they see us all the time caring for their relative, but nobody says anything. We are 24, 30 and more hours with the patient. For me, it'd be strange! [..] However, the family doesn't realise why we are there. More than once they have said 'hey, such a long shift', and we replied 'eh, yes' [. . .]

Davis [49] described functional uncertainty to explain how clinicians manage the information towards patients and families when the disclosure of information is considered problematic and could provoke emotional turmoil, effort or time. In other words, organ donor coordinators use functional uncertainty to deal with challenging communication, which has been described due to its ethical and moral implications in similar studies [53–55]. Nurse coordinators trust their feelings and experience of perceiving the family to inform, support them to what extent and when. It other words, *Reading the family* defines functional uncertainty, and critically what to say, how to say it and when. Although approaching families from the beginning of the experience and building an early time-compressed intimacy allows supporting constantly families, it also requires flexible social norm related to communication process. One physician reflects.

Physician 6:. . .I came from a school where was the physician who will exclusively deliver the news and speak to families in all cases. Yet here [A&E unit], the communication process can be difficult due to workload, speed of admissions; it is very stressful and sometimes the service is at point of collapse, so communication is not always as it should be [..] The organ donation team helps that process and leads the communication with families of potential organ donors [..] My reflexions after being part of the communication process led by the team, it is the need of a multidisciplinary approach.

Flexible social norms related to delivering bad news and leading the communication process are also clearly associated with a belief system of clinicians that may help or hinder the care for families in this context [37]. Adopting this flexible approach, providing clear and constant communication and support to families, was also highlighted by interviewed families. Mario, who lost his father, recalls the experience in these terms.

Mario: Miss (coordinator's name) was with us since Saturday night when my father got hospitalised until the day after. I mean, we got the whole process with her. She was with us. . .She let us see him, to be there with him. And . . .she cared very much for us.

Very well supported. She. . .yes, she was very kind. Everything that we needed she was keen to do it immediately! And after, she let us stay with him in the morning, and after the process was done. . .To be the last moment with him.

Families who experienced *Reading the family* directly were more likely to remember the process with details, and coordinators' name, as Mario mentioned above. Family members acknowledged the support received by nurse coordinators, and consistently expressed gratefulness through body language and words. Family members narratives were characterised by closeness and familiarity, demonstrating a developed time-compressed intimacy. Many of them recalled having the contact number in their cell phones, which make them feel in touch with nurse coordinators and the hospital. The positive response of families towards an early time-compressed intimacy might reinforce nurse coordinators beliefs regarding the essential role of looking after families throughout the process, regardless the outcome of organ donation authorisation. Moreover, creating a relationship with the family permits nurse coordinators to feel close to the families, accurately assessing their emotions and needs. One nurse coordinator reflects on how families set up the flow of the approach.

Nurse coordinator 5: I think that also influences how the family is reacting.

LA: What do you mean?

Nurse coordinator 5: For example, in the first approaching that I made. I think that if it would've been the same situation as the second approach, I would've cried.

It was the brother and the mother, and they were calm. . .it was a suicide (whispering). Then, they said. . .they respected their will. And it was so, so! The family understood so well that it wasn't a major emotional effort for us to try to. . .to keep them calm or make them comprehend the brainstem death. I didn't get the emotional burden; instead, they were so serene. . . that they calmed me down as well!

Nurse coordinators focus on supporting the family to cope with their loss experience [31] but also the process helps nurse coordinators to develop edgework emotion management, managing their emotions and situation [26]. *Reading the family* defines the means of communication and how nurse coordinators support the family and negotiate organ donation authorisation.

**Negotiating organ donation conversations using Reading the family.**   *Reading the family*, as the process to explain nurse-family interaction, guides the negotiation process by emotional dialogue and mutual influence of emotions, balancing power dynamics. Orøy et al. [46] interviewed 32 healthcare professionals when approaching families for organ donation in Norway, reporting similar findings. The authors refer to the negotiating as a unique situation in which "some nurses stated that the situation was determined by the people involved, healthcare professionals and relatives alike" [p.207]. A nurse said, "[t]he question is what do we do and how do what we do. That's also the challenge. Hence, we cannot have procedures" [p.207]. In this study, nurse coordinators crafted the family approach using *Reading the family* as a core process. They reflected.

Nurse coordinator 3: [..] The family is in shock due to the death. We ask some questions, how was she/he? What did she/he do? [the potential organ donor] We make the family

change their emotions and feelings to be able to talk. [..] After there is a brief silence. . .and the family says, what do we do now? And we raise the discussion. There is a possibility to help someone else [..]

Nurse coordinator 1: I think that is something about. . .

Nurse coordinators 3: the feeling

Nurse coordinator 1: you feel it

Nurse coordinator 3: yes, it is like. . .

Nurse coordinator 1: just the right moment to ask

LA: how come? Have you thought about it?

Nurse coordinator 2: I mean, every family is just very different [..]

Nurse coordinator 3: Although we got a sort of structure, it is case by case. We observe everything, how is the family, their emotions, their feelings. It is not only the physicality; it is also how they speak. . .

(Focus Group 6, C1)

By *Reading the family*, nurse coordinators negotiate organ donation conversations as a relational process in which emotions set the tone of how, when and what to say. Emotions are constructed and managed to share meanings within reciprocal and continuous interactions [34, 52]. One observed approaching involved a young male who died due to a traumatic brain injury, the neurologist informed the death and told the family about the need to discuss the following steps. The field note captured.

The coordinator and I were waiting for the family to say something. The coordinator looked at them, waiting. The family was offered to stay in the room if they wanted, but the mother was silent, shocked. Her partner wanted to go away. He said, 'I wanna go. I wanna go. Let's go, let's go'. The coordinator said, 'we're sorry, we're really sorry', and offered them to facilitate the entrance to the unit to say goodbye. The family agreed. On our way out, the mother walked, unstable. We held her and helped her to walk. We went with them to the hospital entrance to meet the extended family who was waiting outside.

We waited for a while, looking at how the family cried together. The coordinator approached another family member, the sister-in-law of the mother, and explained the needed to talk to the parents again, but when 'they felt calmer'.

(Field note, November 2017, Approaching 1, C1)

When *Reading the families* is performed throughout the process of hospitalisation, coordinators can sense how the family has processed the information and able to manage family emotions accordingly. In this example, coordinators provided not only physical and emotional support, but also more time and space. The coordinator also used functional uncertainty [49] to talk again when 'they felt calmer'. They sensed that the family was not able to cope with anything else at that moment. Coordinators encourage the expression of feelings and support the family as the priority. The following encounter with this family was two hours later when they went to the host room again. Although they did not donate, the family had time to say goodbye, received information regarding body discharge and the examinations post-mortem due to the death in the context of violence.

*Reading the family* guides to carefully negotiate organ donation conversations. During another approach, the encounter included the husband, son, daughter and the sister of the potential organ donor, and the following field note describe the negotiation process.

[..] the coordinator continued describing the process of organ retrieval. The coordinator spoke slowly and looked at family reactions. Considerations the impossibility to know the recipients were also discuss due to current legislation. The only information that could be provided is the number of recipients beneficiated. Additionally, the negotiation of tissue donation started, naming cornea, bones, and skin [..] The coordinator made a pause. The family was silent listening. The coordinator continued explaining that the cornea can help people of the same hospital to recover their sight [..] Both siblings looked at each other and agreed, nodding. The father said, 'yes, it is ok'. (Field note, December 2017, Approaching 4, C1)

The process of negotiating organ donation conversations for organ donation authorisation is carefully managed and carried out by skilled nurse coordinators. The observation above described the careful assessment of family reactions to keep going. It was evident while observing that the nurse coordinator evaluated family's body language, to read family' emotions, looking for hesitation, distress and uneasiness. Nurse coordinators are aware that the manipulation of the body is one of the main concerns for families and reasons to refuse organ donation [48, 56], and the reason why the coordinator paused to assess the family.

*Reading the family* describes the skilled but often hidden process of nurse coordinator's ability to access, assess and manage family's emotions at the time of the loss to start the negotiation process of organ donation conversations. *Reading family* draws on Brinkmann's [57] definition of emotions as "a kind of practical wisdom and an ability to understand other people and their situations" [p.42]. This study argues that *Reading the family* has been described in similar studies yet not fully articulated [46, 58]. Recent studies have highlighted the importance of the rapport between coordinators and families to discuss organ donation [59, 60] and few studies have associated a longer approaching period with increased organ donation consent rates [61]. Perhaps, referring to an early time-compressed intimacy that could facilitate trust to better support families in the process of dying and potentially better results when negotiating organ donation authorisation.

*Reading the family* as a social process was evident during the interactions, yet often overlooked, becoming invisible. Emotions are embodied and interactional agents that are constructed and managed to share meanings within reciprocal and continuous interactions [34, 52]. Danet et al. [58] interviewed 22 transplant coordinators in Spain and the participants said. "For me, it is always novel, and each interview is an outpouring of emotions [p.77]", ".. each interview is different and there is always something that you have not confronted [p.78]". Spanish data expressed not only the awareness of emotions in approaching families but also how coordinators craft their approach to each unique encounter. This study argues that approaching families involved a relational emotional dialogue between coordinators and families; called *Reading the family*.

## Limitations

The limitations to study's cultural context and thus transferability of the grounded theory needs to be considered. The study explored the experiences of clinicians and families for organ donation after brainstem death (DBD) only, and transferability of this grounded theory into different cultural and clinical context therefore requires further research. The constructivist

stance also acknowledges the co-construction of data between researcher and participants, and a detail account of these interactions and author's positionality was provided.

## Conclusion

This study developed a constructivist grounded theory to explain the process of approaching families to negotiate organ donation conversations for organ authorisation in Chile from the perspective of clinicians and families [43]. *Reading the family* describes the skilled and often hidden process of assessing and managing family's emotions to carry out difficult conversations. In this study, it defines communication, care and support provided to negotiating organ donation authorisation. The core process can be indirect and direct, which critically depend on the clinicians set of beliefs [37] and communication processes. Findings demonstrate how nurse-family relationships and interaction are mediated by beliefs and emotions in the context of negotiating organ donation authorisation. Data and evidence demonstrated that every family encounter constitutes a unique experience [46, 58], and a result standardised guidelines could risk simplifying a complex human interaction within cultural and social norms. Evidence suggests that despite regulations and guidelines, healthcare professionals craft the approach In Chile, organ donation nurses approach families as a family unit, which has also reported in related studies [62], acknowledging cultural beliefs of death as a family affair and supporting families at the end of life. However, this theory could help the development of emotion management training for organ donation coordinators worldwide and potentially other healthcare professionals.

*Reading the family* as a process has been described in similar studies yet not fully analysed, supporting this conceptualisation. This theory was developed to understand nurse-family interactions to navigate difficult conversations within the organ donation process, but it could also guide end-of-life conversations in other contexts and situations. Equally its potential beyond nursing, and therefore, further research is needed and suggested to assess the feasibility of its application.

## Supporting information

**S1 Checklist. COREQ checklist.**
(PDF)

**S1 Appendix. Data collection tools.**
(DOCX)

## Acknowledgments

My deepest gratitude to all participants, healthcare professionals and families who kindly consented to be part of the study and shared their valuable experiences. I also thank my amazing supervisors and colleagues Dr Susanne Kean and Dr Jennifer Tocher for their guidance and support throughout my PhD.

## Author Contributions

**Conceptualization:** Lissette Avilés.

**Data curation:** Lissette Avilés.

**Formal analysis:** Lissette Avilés.

**Funding acquisition:** Lissette Avilés.

**Investigation:** Lissette Avilés.

**Methodology:** Lissette Avilés.

**Project administration:** Lissette Avilés.

**Resources:** Lissette Avilés.

**Software:** Lissette Avilés.

**Visualization:** Lissette Avilés.

**Writing – original draft:** Lissette Avilés.

**Writing – review & editing:** Lissette Avilés.

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
