## [Decision Letter · Decision Letter 0]

21 Jun 2024

PONE-D-24-14795Reading the family: a constructivist grounded theory on organ donation conversationsPLOS ONE

Dear Dr. Avilés,

Thank you for submitting your manuscript to PLOS ONE. After careful consideration, we feel that it has merit and might meet PLOS ONE’s publication criteria after minor revisions. Therefore, we invite you to submit a revised version of the manuscript that addresses the points raised during the review process. The reviewers ask questions and make valuable suggestions that should help you improve your manuscript. You do not need to respond to ALL comments by making changes to the text of the article. See them instead as opportunities for improvement. However, I ask you to pay special attention to the reviewers’ comments regarding the methods and to address their requests for clarification. Please submit your revised manuscript by Aug 05 2024 11:59PM. If you will need more time than this to complete your revisions, please reply to this message or contact the journal office at plosone@plos.org. Please include the following items when submitting your revised manuscript:A rebuttal letter that responds to each point raised by the academic editor and reviewer(s). You should upload this letter as a separate file labeled 'Response to Reviewers'.A marked-up copy of your manuscript that highlights changes made to the original version. You should upload this as a separate file labeled 'Revised Manuscript with Track Changes'.An unmarked version of your revised paper without tracked changes. You should upload this as a separate file labeled 'Manuscript'.If applicable, we recommend that you deposit your laboratory protocols in protocols.io to enhance the reproducibility of your results. Protocols.io assigns your protocol its own identifier (DOI) so that it can be cited independently in the future. For instructions see: https://journals.plos.org/plosone/s/submission-guidelines#loc-laboratory-protocols. Additionally, PLOS ONE offers an option for publishing peer-reviewed Lab Protocol articles, which describe protocols hosted on protocols.io. Read more information on sharing protocols at https://plos.org/protocols?utm_medium=editorial-email&utm_source=authorletters&utm_campaign=protocols.

We look forward to receiving your revised manuscript.

Kind regards,

Alberto Molina Pérez, Ph.D.

Academic Editor

PLOS ONE

 [The research was funded by the Agencia Nacional de Investigacion y Desarrollo, Becas Chile - PhD scholarship, Chile (Folio N. 72160141).].  

3. We noted in your submission details that a portion of your manuscript may have been presented or published elsewhere. [Yes, table 1 and 2 have been already published as part of the data set of the study and acknowledged within the paper.] Please clarify whether this publication was peer-reviewed and formally published. If this work was previously peer-reviewed and published, in the cover letter please provide the reason that this work does not constitute dual publication and should be included in the current manuscript.

Reviewers' comments:

Reviewer's Responses to Questions

**Comments to the Author**

1. Is the manuscript technically sound, and do the data support the conclusions?

Reviewer #1: Yes

Reviewer #2: Yes

2. Has the statistical analysis been performed appropriately and rigorously? 

Reviewer #1: N/A

Reviewer #2: N/A

3. Have the authors made all data underlying the findings in their manuscript fully available?

Reviewer #1: No

Reviewer #2: Yes

4. Is the manuscript presented in an intelligible fashion and written in standard English?

Reviewer #1: Yes

Reviewer #2: Yes

5. Review Comments to the Author

Reviewer #1: Dear author,

you have conducted a very comprehensive empirical study that provides a detailed insight into the constructivist approach of: Reading the family. In my view, your empirical study is very well presented. Your qualitative empirical results are clearly to understand. There is a good balance between analytical results and empirical anchor examples.

I have some suggestions for the presentation and better comprehensibility of the empirical data.

1. Provide more detailed information about your interviews and focus groups. Create a table with the composition of the individual focus groups to make it easier to understand the composition. It would also be good to attach both the discussion guide for the focus groups and the interview guide for the interviews as supplementary material.

2. Explain in more detail how the analysis process was carried out, e.g. with a cathegory table, which could be added as an attachment.

2. In the introduction, you talk about different cultures in which the topic of the organ donation process has already been investigated. In my view, you are referring more to nations (and there organ transplantation system). Culture as a term seems misleading to me, as it also refers to different organ donation practices.What seems more important to me, as you also mention in your work, are different family cultures. For example, how families see themselves as a group and how this determines the way they deal with problems. In my opinion, it would be useful to differentiate this point even more in the introduction.

Kind regards

Reviewer #2: The aim of this article is to find out what the encounter between professionals and families is like at the time of requesting a donation. It develops a constructivist grounded theory based on field material collected, as well as interviews and focus groups with different agents involved in the process.

General comment: Thank you for the work, it seems to me a very interesting and relevant topic, I have learned a lot reading this work. I would like to make some considerations (some questions and ask for some clarifications):

Regarding the introduction

Contextual information on the specific training that nurse coordinators have in Chile, do they receive courses on communication skills? What are these courses like? In Spain, for example, the "Alicante Model" is used, which has also been exported to other countries (Gómez, P., & de Santiago, C. (2008). The family interview. Technique and results. In El modelo español de Coordinación y Trasplantes Aula Médica). Are there data on refusal rates?

Regarding the methodology

I suppose the extension is a limit, but I would like you to further develop some points of the methodology that I consider very interesting. The methodology is very original, in other contexts participant observation has been proposed as a useful methodology to investigate the phenomenon of donation but very difficult to carry out due to the ethical challenges it poses (López, M. V. M. M., Nieto, E. M. M. M., & Piqueras, M. C. (2022). Permission to investigate? Reflections on the ethical requirements of participant observation in the context of the family organ procurement interview. RECERCA. Revista de Pensament i Anàlisi, 27(2), https://doi.org/10.6035/recerca.6148).

Could you please clarify the limitations, difficulties you had in carrying it out and the procedure followed?

Was the analysis in C2 done at a later stage? Clarify whether data collection was done in phases or simultaneously. A graph would be nice to know the sequence of data collection. Table 2 pertains to the socio-demographic profile of the participants but I would like to know who was interviewed in depth, or who participated in the focus groups. Were the focus groups mixed between professionals and families?

With regard to the results

As the authors say, reading emotions is related to the cultural context, decision making is a family issue. The theory has two dimensions, indirect and direct, which refers to when and how nurses read the emotions of families for and support the decision making process of bereaved families. Is it a capacity or a skill? Is it trained with training? Exclusive to nursing? What makes nursing special to this theory? Could the ability to "read families" be extended to other professionals or is it specific to nursing competences? Do you think it is relevant for a feminised profession like nursing to take on this role? Were the coordinators mostly women? reflection on gender is needed. Finally, do you think that profession, gender, training and years of experience play a relevant role?

There is a profound debate about whether it is ethically acceptable for the same people who have been caring for the patient to be the transplant coordinators and also make the request to the family (as is the case in Spain), or whether it is preferable for it to be an external figure (as is the case in the UK or Chile). I wonder if the reading of emotions and the family (this theory) would vary if it were not an external figure but the person who has been caring for the patient and family in the ICU. Would the reading of emotions be more profound when one has participated in the process of care in the ICU for both the patient and the family or is it necessary for this reading the distance provided by being an external nurse coordinator?

I think we could go a little deeper into trust, that "click" and that reading of the family cannot be established if there is no clinical relationship built on personal trust. I think it would be good to reflect on this, in case it is useful: Martínez-López, M. V., McLaughlin, L., Molina-Pérez, A., Pabisiak, K., Primc, N., Randhawa, G., Rodríguez-Arias, D., Suárez, J., Wöhlke, S., & Delgado, J. (2023). Mapping trust relationships in organ donation and transplantation: A conceptual model. BMC Medical Ethics, 24(1), 93. https://doi.org/10.1186/s12910-023-00965-2

Congratulations because the verbatims are very well chosen, they have the ability to portray the heartbreaking moment of loss.

The importance of accompaniment for the relatives is pointed out, I think it would also be good to point out the importance of rituals (based on the values and beliefs of the patient and families) and of the farewell, as well as the work of the professionals in facilitating it.

I think there is a typo in the supplementary material: "The other nice preferred their houses" (appendix 2 bottom of page 2). I think they mean "the other nine".

Ethical issues

The presence of a researcher affects the context (given that she is an external agent who is observing). How was this issue managed? Was it taken into consideration that the presence of the researcher could lead to a refusal of the application for donation? What protective measures were taken with the participants? Was time allowed for the interview in order to allow them time to grieve? Was care for the participants taken into account because they were in a situation of special vulnerability? I would like to congratulate the reflection on the researcher's own care, which is usually not taken into account and is of utmost importance.

6. PLOS authors have the option to publish the peer review history of their article (what does this mean?). If published, this will include your full peer review and any attached files.

Reviewer #1: **Yes: **Sabine Wöhlke

Reviewer #2: **Yes: **María Victoria Martínez-López

---

## [Author Response · Author response to Decision Letter 0]

26 Jul 2024

I've responded to all reviewers comments on attached file

---

## [Decision Letter · Decision Letter 1]

8 Oct 2024

Reading the family: a constructivist grounded theory on organ donation conversations

PONE-D-24-14795R1

Dear Dr. Avilés,

We’re pleased to inform you that your manuscript has been judged scientifically suitable for publication and will be formally accepted for publication once it meets all outstanding technical requirements.

Kind regards,

Michal Mahat-Shamir, Ph.D.

Academic Editor

PLOS ONE

Additional Editor Comments (optional):

This paper presents a robust exploration of a highly significant topic, employing sound and well-structured methodology. In my view, the authors have effectively addressed all the concerns raised during the initial round of reviews, demonstrating careful consideration and thoughtful revisions. As a result, I find that the paper meets the necessary criteria for publication and is a valuable contribution to the field.

Reviewers' comments:

Reviewer's Responses to Questions

**Comments to the Author**

1. If the authors have adequately addressed your comments raised in a previous round of review and you feel that this manuscript is now acceptable for publication, you may indicate that here to bypass the “Comments to the Author” section, enter your conflict of interest statement in the “Confidential to Editor” section, and submit your "Accept" recommendation.

Reviewer #2: All comments have been addressed

Reviewer #3: (No Response)

2. Is the manuscript technically sound, and do the data support the conclusions?

Reviewer #2: Yes

Reviewer #3: Yes

3. Has the statistical analysis been performed appropriately and rigorously? 

Reviewer #2: I Don't Know

Reviewer #3: N/A

4. Have the authors made all data underlying the findings in their manuscript fully available?

Reviewer #2: Yes

Reviewer #3: No

5. Is the manuscript presented in an intelligible fashion and written in standard English?

Reviewer #2: Yes

Reviewer #3: No

6. Review Comments to the Author

Reviewer #2: In my opinion they have answered all my questions and addressed them as far as possible.

I recommend acceptance of this manuscript for publication in the journal.

Thank you for giving me the opportunity to review.

Reviewer #3: This paper presents research that is very important for the development of evidenced-based practice.

It is inovative and the data presented support the conclusions.

There are some aspects of the paper that require improvement before publicaion.

First and foremost the mistakes in grammar and syntax must be corrected. I made corrections as I read the paper until line246 (see below). At that point I gave up, since I was starting to feel like a proofreader. You must send the paper for English language editing to a professional who is a native English speaker/writer.

Second, although I very much appreciated the integration of the data analysis and interpretation with pertinent research, I believe that the literature review in the introduction is too scant. Please report on the major findings on the research topic.

Third, some of the concepts you discuss or develop are not clearly defined. (See below)

Fourth, please provide information about the cultural context in Chile re: beliefs about the body after death, attitudes towards health care professionals, and expression of emotion.

Below are detailed comments about the contents of the paper.

Line 75: What do you mean by "family care" in this context?

Line 94: Data analysis is not conducted face to face.

Line 96: The porposive and theoretical sampling and the shift from one to the other should be explained.

Line 121: Who was included in the family subsystem?

Line 123: Focus groups with professionals or families?

Line 125: What does not returning the transcripts to participants have to do with grounded theory analysis?

Line 133: meaning construction?

Line 152: Solomon's theory of emotions needs to be explained further.

Line 154: Are there skills and strategies for feeling emotions?

Line 155: "Early in the study"--at what stage?

Table 1: "9 family members" from how many family subsystems?

What are family encounters (as opposed to family approaches)?

Are the donor authorisations the endpoint of the family approaches? If so, they should be listed directly

after.

Line 168: Please define resonance in this context.

Line 205: Again, who is included in the "family unit"?

Line 265: It seems that you are proposing here that the emotional response of the nurse is spontaneous and leads to a more calculated joing with and assessment of the family. Perhaps this should be articulated.

How is "reading" different from assessing? the emotional component on the part of the nurses?

Line 285:How is family "readiness" determned?

Line 317: Define "edgework emotion management."

Line 478: What is A&E?

Line 484: Define functional uncertainty

Line 490: I don't think that reading the family defines functional uncertainty, but rather leads to it or is a way of dealing with it.

Line 493: "flexible social norms" Please explain in this context.

Line 581: "Emotions are constructed." What does this mean?

Line 642: These exact words were written earlier on in the paper.

Grammatical errors (up until line 246):

l. 62: This sentence needs to be rewritten.

l. 64: Change "to" to "of".

L. 68: Change to "consent obtained by physicians."

l. 69: Rewrite this sentence.

l. 71: Remove the word "and".

l. 75: Change "internationally" to "in different countries."

l. 76: Remoce the word "even."

l. 77: Add comma after the word "death".

l. 83: Change "approach" to "approached".

l. 84: Add the word "method" or "approach" after the word "theory".

l. 85: Repalce "comparative methods" with "constant compariso.""

l.86: Consider changing "abduction" to "abductive reasoning."

l. 93: "Participant" should appear in the singular form.

l. 101: The sentence about adult donors and non-donors is not clear at all.

l. 108: Incomplete sentence

l. 137: Is there a word missing after "groups"?

l.149: The meaning of this sentence is unclear.

l. 150: Correct the mistakes in syntax.

l. 155: Incomplete sentence

l. 158: Add the word "that" before "the".

l. 178: Add the word "theory" after "grounded".

l. 179 Change "logic" to "logicaj",

l. 209: Insert the word "as" before the word "cnflicted"/

l. 226: Correct margin.

l. 229: Incorrect syntax.

l. 230: I think you mean "sensitive", not "sensible"/

l. 231: Correct the grammar and insert commas

l. 234: Change to "two different approaches."

l. 246: Change "in" to "on".

7. PLOS authors have the option to publish the peer review history of their article (what does this mean?). If published, this will include your full peer review and any attached files.

Reviewer #2: **Yes: **María Victoria Martinez-Lopez

Reviewer #3: **Yes: **Chaya Possick

---

## [Editor Report · Acceptance letter]

10 Oct 2024

PONE-D-24-14795R1 

PLOS ONE

Dear Dr. Avilés, 

I'm pleased to inform you that your manuscript has been deemed suitable for publication in PLOS ONE. Congratulations! Your manuscript is now being handed over to our production team.

Kind regards, 

on behalf of

Prof. Michal Mahat-Shamir 

Academic Editor

PLOS ONE